# Omecamtiv mecarbil lowers the contractile deficit in a mouse model of nebulin-based nemaline myopathy

**Johan Lindqvist**☯**, Eun-Jeong Lee**☯**, Esmat Karimi, Justin Kolb, Henk Granzier**[ID]*

Department of Cellular and Molecular Medicine, University of Arizona, Tucson, Arizona, United States of America

☯ These authors contributed equally to this work.
* granzier@email.arizona.edu

**Data Availability Statement:** All data are contained within the paper and its Supporting Information files.

## Abstract

Nemaline myopathy (NEM) is a congenital neuromuscular disorder primarily caused by nebulin gene (NEB) mutations. NEM is characterized by muscle weakness for which currently no treatments exist. In NEM patients a predominance of type I fibers has been found. Thus, therapeutic options targeting type I fibers could be highly beneficial for NEM patients. Because type I muscle fibers express the same myosin isoform as cardiac muscle (Myh7), the effect of omecamtiv mecarbil (OM), a small molecule activator of Myh7, was studied in a nebulin-based NEM mouse model (*Neb* cKO). Skinned single fibers were activated by exogenous calcium and force was measured at a wide range of calcium concentrations. Maximal specific force of type I fibers was much less in fibers from *Neb* cKO animals and calcium sensitivity of permeabilized single fibers was reduced ($pCa_{50}$ 6.12 ±0.08 (cKO) vs 6.36 ±0.08 (CON)). OM increased the calcium sensitivity of type I single muscle fibers. The greatest effect occurred in type I fibers from *Neb* cKO muscle where OM restored the calcium sensitivity to that of the control type I fibers. Forces at submaximal activation levels (pCa 6.0–6.5) were significantly increased in *Neb* cKO fibers (~50%) but remained below that of control fibers. OM also increased isometric force and power during isotonic shortening of intact whole soleus muscle of *Neb* cKO mice, with the largest effects at physiological stimulation frequencies. We conclude that OM has the potential to improve the quality of life of NEM patients by increasing the force of type I fibers at submaximal activation levels.

## Introduction

Nebulin is a giant filamentous protein, located in the skeletal muscle sarcomere[1] where it winds around the actin filament, from the Z-disk to near the pointed end of the actin filament [2]. Nebulin is believed to play an important role in muscle contraction[3–5], it maintains myofibrillar alignment[6] and functions as a thin filament stabilizer that regulates thin filament length[7]. Nebulin deficiency due to mutations in the nebulin gene (NEB) is an important cause of Nemaline Myopathy (NEM)[8–11]. Although the phenotypes of nebulin-related

**Funding:** Research reported in this publication was supported by the National Institute of Arthritis and Musculoskeletal and Skin Diseases of the National Institutes of Health under Award Number R01AR053897 (HG), by a grant from A Foundation for Building Strength (JL), and by a grant from the Leducq Foundation, 13CVD04 (HG).

**Competing interests:** The authors have declared that no competing interests exist.

NEM patients are variable, a common functional feature is skeletal muscle weakness [12–17]. No treatments for NEM exist to improve muscle function. In NEM patients a predominance of type I fibers has been found [13, 16]. Thus, therapeutic options targeting type I fibers would be highly beneficial for NEM patients.

Omecamtiv mecarbil (OM) is a selective small-molecule activator of cardiac myosin (Myh7 or β-MHC) that was developed as a treatment for heart failure [18, 19]. Single molecule studies have shown recently that OM suppresses the size of the working stroke of myosin and prolongs the duration of its attachment to actin [20]. It has been proposed that OM-inhibited myosin heads cooperatively activate the thin filament at submaximal activation levels, recruiting OM-free myosin heads and increasing force [20].

Although OM has been developed for heart failure patients, cardiac myosin (Myh7 or β-Myh) is identical to the type I myosin expressed by slow skeletal muscles[21] suggesting that OM might also be effective in skeletal muscle. Furthermore, type I fibers are the dominant fiber type in most human muscles [22, 23]. The percent type I fibers in healthy humans is >60% for a wide range of muscle types [23] and examples of type I rich muscles in the mouse are the piriformis (~60%), the quadratus femoris (~70%)[24], and the soleus (~50%, Fig 1). Furthermore, studies have shown that in both mouse models of NEM and in NEM patients there is an additional shift towards type I fibers [12–15, 25]. Thus, OM might be effective in increasing skeletal muscle force, particularly in NEM patients. We investigated whether OM has a beneficial effect on slow skeletal muscle and used a conditional nebulin knockout mouse model (*Neb* cKO) which expresses low levels of nebulin and has severe weakness in peripheral and respiratory muscles[25], phenocopying NEM[9, 16, 25, 26]. Additionally, there is a fiber type I predominance [25], as in NEM patients. Hence, we tested the hypothesis that OM ameliorates the force deficit in *Neb* cKO mice.

## Materials and methods

The conditional nebulin knockout (*Neb* cKO) mouse was previously described [25]. Mice that were positive for MCK-Cre and homozygous for the floxed nebulin allele were nebulin deficient and are referred to as *Neb* cKO. Littermate controls (CON) were WT for nebulin[25]. Studies were performed on membrane-permeabilized soleus muscle fibers and intact whole soleus muscles. Six months old *Neb* cKO (189±3 days old) and CON (186±1 days old) mice were used for this study. In this work we focused on the fiber type I rich soleus muscle. The weight of the soleus muscle was 7.7 ± 0.5 mg (n = 20) in CON and 11.6 ± 0.7 mg (n = 25) in Neb cKO mice and the physiological cross-sectional area at mid-belly was 0.58 ± 0.03 mm$^2$ (CON) and 0.91 ± 0.05 mm$^2$ (*Neb* cKO). This is similar to our previous study [25] in which it was found that most muscles in the *Neb* cKO mice are atrophied, with as exception the soleus muscle that hypertrophies. Furthermore, it is known from previous work that the fiber cross-sectional area of type I fibers in the soleus is reduced significantly in the *Neb* cKO while the number of type I fibers is greatly increased [25]. All experiments were in accordance with the United States Public Health Service's Policy on Humane Care and Use of Laboratory Animals and approved by the University of Arizona Institutional Animal Care and Use Committee.

### Omecamtiv mecarbil

Omecamtiv mecarbil (OM) was purchased from Selleckchem (Houston, TX) and was dissolved in dimethylsulfoxide (DMSO) to make a stock solution as instructed by the manufacturer. OM stock solution was then added to the experimental solution to prepare the final desired concentration of OM. In most studies, we used a low OM dose (0.1 μM) to limit effects on cardiac muscle that occur mainly at OM concentration > 0.3 μM [18, 19, 27–30].

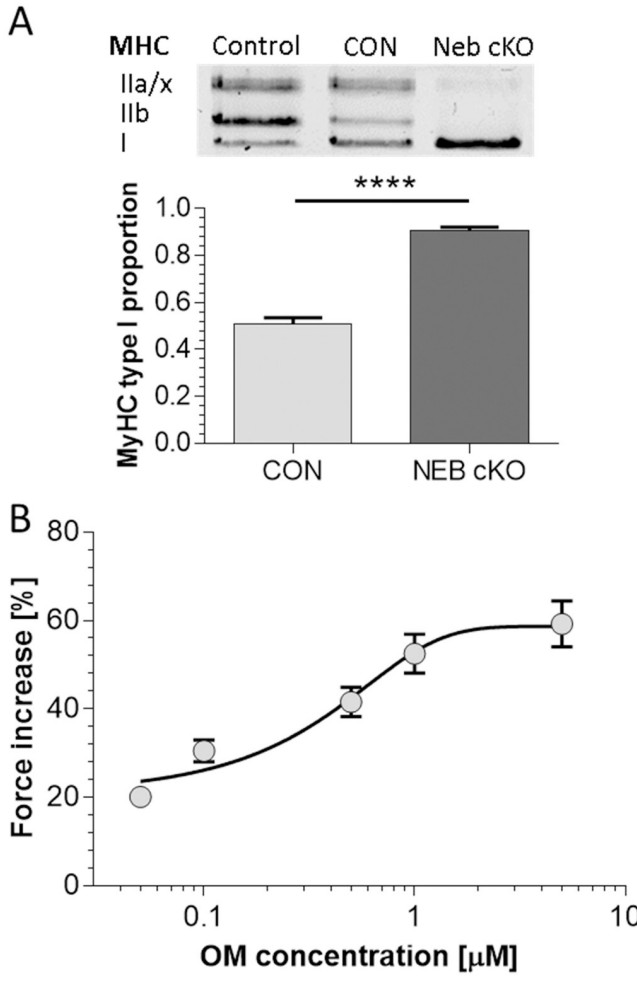

**Fig 1. Fiber type composition of *Neb* cKO muscle and dose-response curve of OM.** (A) The proportion of type I myosin heavy chain (MHC) isoforms. (Top) Typical MHC-gel of soleus muscle of six months old CON and cKO mice. Control muscle (left lane) is a mixture of 40 days old TC and Sol muscle to reveal all MHC isoforms. In CON, MHC type I and IIa/x isoforms are present, while only the MHC type I isoform is clearly detectable in Neb cKO muscle. Bottom: The proportion of type I MHC is significantly greater (****: p<0.0001) in Neb cKO (n = 25) than CON (n = 20) mice. (B) Dose-response of OM on soleus muscle from *Neb* cKO mice (n = 6). Forces were measured at a 20Hz stimulation frequency. Relative increases in force from vehicle are shown. Relative increases in force beyond vehicle are shown. Results follow a dose-response curve with $EC_{50}$ of 0.47±0.08 µM (based on fitting results from each individual muscle). Values are mean ± SEM.

## Skinned muscle fiber mechanics

**Experimental procedure.** The procedures for skinned muscle mechanics were as described previously[31]. Briefly, soleus muscles were skinned overnight at $\sim 4°C$ in relaxing solution (in mM: 40 BES, 10 EGTA, 6.56 MgCl$_2$, 5.88 NaATP, 1 DTT, 46.35 K-propionate, 15 creatine phosphate, Ionic strength 180 mM, pH 7.0 at 20°C) containing 1% (w/v) Triton X-100 and protease inhibitors (in mM: 0.01 E64, 0.04 leupeptin and 0.5 PMSF). Muscles were then washed thoroughly with relaxing solution and stored in 50% glycerol/relaxing solution at −20°C[32]. Either fiber bundles (Fig 2) or single fibers (Figs 3 and 4) were dissected and mounted using aluminum T clips between a length motor (ASI 322C, Aurora Scientific Inc.),

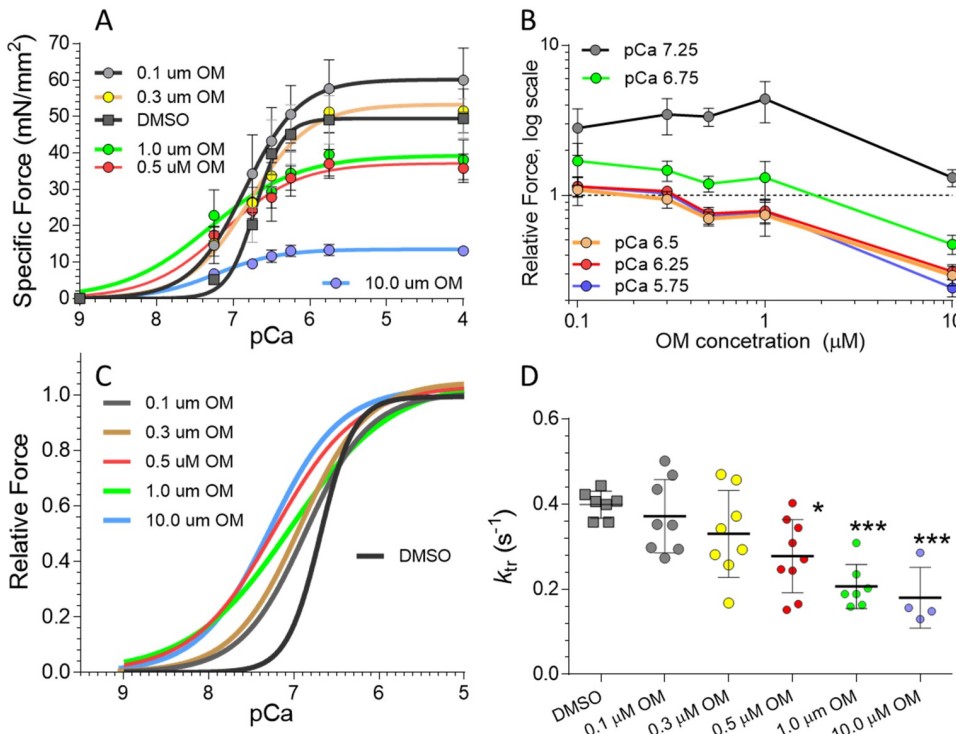

**Fig 2. Effect of OM on force-pCa relations of permeabilized fiber bundles from Neb cKO mice.** A) Force-pCa curves in 0, 0.1, 0.3, 0.5, 1.0 and 10.0 μM OM. OM affects specific force in a dose-and activation-dependent manner. B) Force in OM relative to 0.0 OM at 5 pCa values. The two lowest OM concentrations (0.1 and 0.3 μM OM) increase force at low activation levels with minimal effects at high activation. At 0.5, 1.0, and 10 μM OM forces are minimally increased at low activation level and forces at high activation levels are depressed. C) Force-pCa curves normalized to their maximal force (pCa 4.0) are left shifted in OM (increased pCa50) and less steep (reduced Hill coefficient). D) Ktr values (pCa 6.75) at a range of OM concentrations. At high OM doses, Ktr is reduced. A and B shows mean ± SEM and C shows curve fits of data from 7 muscles except for the DMSO (vehicle) and 10 μM OM data that represent 6 muscles each. In D each data point represents one muscle. Asterisks in D denote significant differences between vehicle and OM (1W ANOVA with multiple testing correction); one symbol p<0.05 and three symbols p<0.001.

and a force transducer element (ASI 403A, Aurora Scientific Inc.) in a skinned fiber apparatus (ASI 802D, Aurora Scientific Inc.). Sarcomere length was set in passive fibers to 2.4 μm using a high-speed camera and video-based sarcomere length software (ASI 900B, Aurora Scientific Inc.); in a small subset of control fibers we measured sarcomere length and found no difference in sarcomere shortening between OM treatment vs. vehicle (DMSO). Muscle bundles/fibers were activated in pCa (pCa = −log([Ca$^{2+}$]) 4.0 activating solution (in mM: 40 BES, 10 CaCO$_3$ EGTA, 6.29 MgCl$_2$, 6.12 Na-ATP, 1 DTT, 45.3 potassium-propionate, 15 creatine phosphate, Ionic strength 180 mM, pH 7.0 at 20˚C) and protease inhibitors. Fiber width and depth (built-in prisms allow for side view of fibers and measurement of depth) were measured at three points along the fiber, and the cross-sectional area (CSA) was calculated assuming an elliptical cross-section. The CSA of type I fibers was on average 1300 μm$^2$ for CON and 1183 μm$^2$ for *Neb* cKO fibers. Specific force was expressed as force per CSA (mN/mm$^2$) and used for comparison of force between groups. In a separate series of experiments we also used skinned cardiac muscle strips (LV papillary)[33] and performed OM dose-response experiments (pCa 6.0) as explained above for skeletal muscle, except that the sarcomere length to which the passive cardiac muscle was set prior to activation was 2.0 μm.

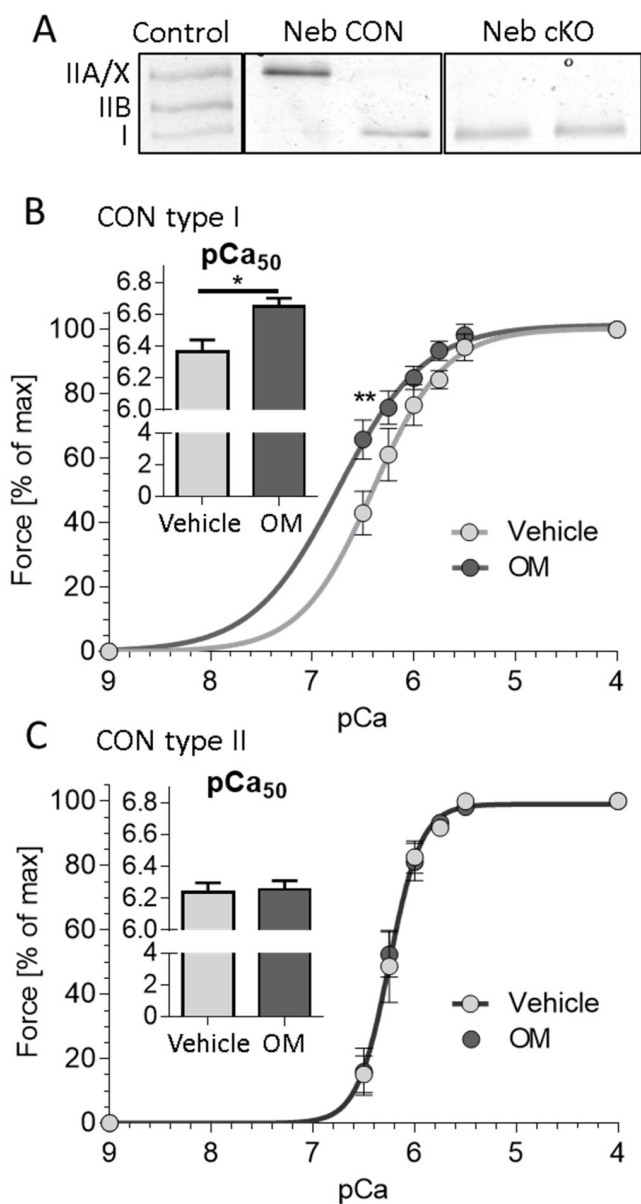

**Fig 3. Effect of OM on active force of single soleus fibers from CON mice.** A) Representative images of silver-stained gel with single fiber lysates loaded. Control is a mix of soleus and EDL lysates that shows the resolvable MHC isoforms (IIX and IIA integrate into a single band). Neb CON shows one type IIX/A fiber and one type I fiber while Neb cKO displays two type I fibers. (B and C) Relative force-pCa curves for slow type I (B) and fast type II (C) muscle fibers from CON soleus mice. Inset in B: Calcium-sensitivity ($pCa_{50}$) of slow type I fibers with 0.1 μM OM was significantly greater than fibers with vehicle. Values are mean ± SEM. Multiple fibers per animal were studied (typically 5–6 fibers per animal) and their means were calculated. The shown values are the means of all studied animals: 5 mice with total 29 fibers (vehicle) and 5 mice with 22 fibers (OM) for type II fibers and 5 mice with 28 fibers (vehicle) and 5 mice with 25 fibers (OM) for type I fibers. (*$p < 0.05$; **$p < 0.01$ comparison vehicle vs. OM).

## Protocols

Preparations were in relaxing solution and then immersed in pre-activating solution (relaxing solution with a 10-fold lower EGTA concentration), followed by activation with incrementally increased pCa (pCa = −log([$Ca^{2+}$]), ranging from 6.5 to 4.0. Solutions with different pCa were

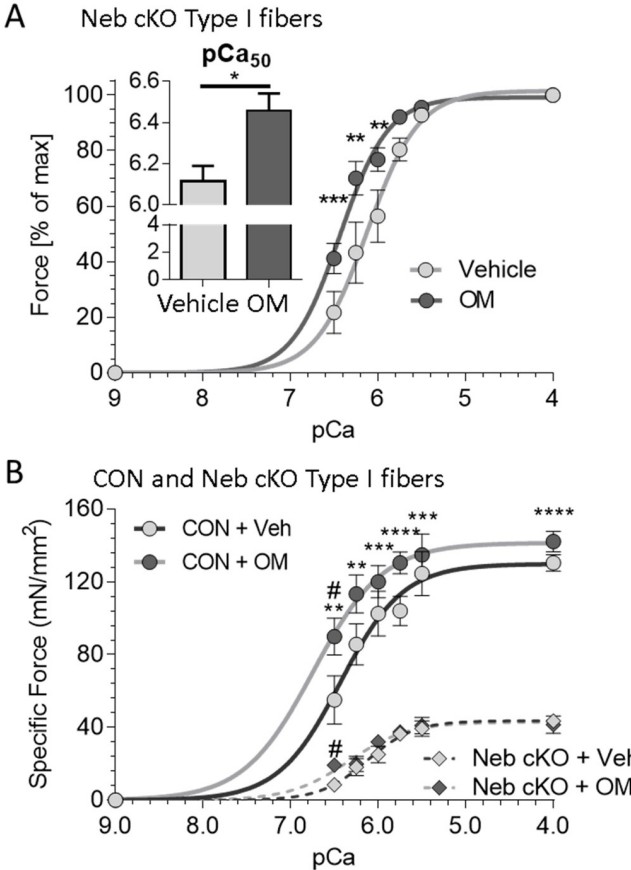

**Fig 4. Effect of OM on active force of single type I fibers of *Neb* cKO and comparison with CON type I fibers.** (A) Relative force-pCa relations of single type I fibers from *Neb* cKO soleus muscle. Relative forces were significantly greater at pCa 6.50, 6.25 and 6.00 with OM in *Neb* cKO fibers. Inset: Calcium sensitivity (pCa$_{50}$) was significantly greater in fibers with OM than with vehicle. (B) Specific force-pCa relations in CON (solid lines) and cKO (dashed lines) in type I fibers with vehicle (Veh) or OM. Force-pCa curves were left shifted with OM relative to vehicle in both genotypes. Values are mean ± SEM. Multiple fibers per animal were studied and their means were calculated. The shown values are the means of 5 mice with total 28 fibers (CON + vehicle), 5 mice with 25 fibers (CON + OM) for type I fibers and 6 mice with 40 fibers (cKO+vehicle) and 5 mice with 28 fibers (cKO+OM). (*p<0.05; **p<0.01; ***p<0.001: ****p<0.0001; comparison CON vs. cKO, #p<0.0001 comparison vehicle vs. OM).

created by mixing varying volumes of relaxing and activation solutions. The obtained force–pCa relation was fit with a Hill equation, providing pCa$_{50}$ (pCa giving 50% of maximal active force) and the Hill coefficient, $n_H$, an index of myofilament cooperativity. Experiments were performed at 20°C. Separate groups of muscle bundles/fibers were used to test the effect of vehicle (DMSO) and OM (in DMSO). Fibers from a total of five CON and five *Neb* cKO mice, with multiple fibers (4–6) per animal were used in fiber experiments. Five to seven soleus bundles from two *Neb* cKO were used for dose-response studies. $k_{tr}$-measurements: The rate of tension redevelopment (ktr) was measured at steady-state force by rapidly shortening (1 ms) the fiber at one end of the fiber resulting in unloaded shortening of the fiber for 20 ms. Remaining bound cross-bridges were detached by rapidly restretching the fiber to initial length and the tension redeveloped [34]. ktr was determined by fitting the rise of force to the following equation (one-phase association curve): F = Fss*(1−ektr*t)+c, where F is force at time t, Fss is steady-state force. After mechanical experiments, the bundles/fibers were stored in SDS-sample buffer for gel electrophoresis and myosin isotyping (see below).

## Intact muscle mechanics

**Muscle experiments.** Mice were anesthetized with isoflurane and sacrificed by cervical dislocation. The soleus muscle was dissected and mounted to the isolated muscle test system (1200A, Aurora Scientific Inc.) that has been described previously[35]. In brief, both tendons were tied with 5–0 silk sutures, and the muscle was attached between the lever arm of dual servomotor-force transducer (300C, Aurora Scientific Inc.) and a fixed hook. The experimental bath was filled with Krebs-Ringer solution containing (in mM) 137 NaCl, 5 KCl, 1 $MgSO_4$, 2 $CaCl_2$, 1 $NaH_2PO_4$, 24 $NaHCO_3$, 11 Glucose with 95% $O_2$ and 5% $CO_2$ supply (pH = 7.4). The temperature during the experiments was 30°C. The muscle was placed between platinum electrodes connected to an electrical stimulator (701C, Aurora Scientific Inc.) for muscle activation. The optimal muscle length ($L_0$) was determined by adjusting overall muscle length until maximal force was generated with a 400ms 20Hz stimulation frequency (pulse duration of 200μs with biphasic polarity). The muscle length at $L_0$ was measured using a digital caliper. At $L_0$, an initial tetanic contraction at 150Hz stimulation frequency was imposed. Measured forces were normalized to the physiological cross-sectional area (PCSA) to obtain specific force (mN/mm$^2$). PCSA (in mm$^2$) was calculated using muscle mass (in g), fiber length (in mm), and muscle density (1.056 g/mm$^3$);PCSA = Muscle mass / (Fiber length × Muscle density). Since muscle fibers of soleus align to their tendon with a pennation angle and do not span the full muscle length, fiber length is estimated from muscle length and pennation angle. As reported in Burkholder et al.[36], the ratio of fiber and muscle length for soleus is 0.72. We used this ratio to calculate the fiber length from the measured muscle length.

**Force-frequency protocol.** To establish the force-frequency relation, active forces at various stimulation frequencies were measured. 14 CON and 14 *Neb* cKO mice were used for the force-frequency protocol. The muscle was stimulated at 1, 5, 10, 20, 30, 40, 60, 100 and 150 Hz and waiting for 30, 30, 60, 90, 90, 120, 120, 180 and 180 s, respectively, in between stimulations. The first force-frequency protocol was performed in the vehicle solution (DMSO), and the experimental solution was then switched to OM solution. The muscle was incubated in OM solution for 15 min with twitch stimulation every minute to monitor changes in force. When the increase in force was stable, the force-frequency protocol was measured.

**Force-velocity protocol.** To determine the force-velocity relation, the load-clamp technique was used as previously described[37]. Shortening velocities were measured during isotonic contraction against loads 85, 70, 55, 40, 25, 15, and 5% of maximal force at 25 Hz of stimulation frequency. Each contraction started out with a 200 ms isometric phase (to reach the plateau force), and then a 150 ms load-clamp was applied. A series of load-clamps was performed in solution with vehicle and the same protocol was then repeated in OM solution. The muscle was incubated in OM solution for 15 min with twitch stimulation every minute before the protocol with OM solution was performed. After completion of each experiment, the sutures and tendons were carefully removed, and the muscle mass was then measured. The muscle was quick-frozen in liquid nitrogen and stored at −80°C for later analysis of fiber type composition.

**Analysis.** The force-frequency curve was fit with a Hill equation to calculate the half-maximal frequency (frequency giving 50% of maximal force increase beyond twitch force) and the Hill coefficient ($n_H$), an index of the steepness of the curve. Shortening velocity (in mm/sec) was measured as a slope of linear portion of the shortening period (~40 ms duration) after force was stable. For each preparation, shortening velocity was normalized by muscle fiber length. Muscle power production was calculated by multiplying the shortening velocity and the applied load. Force-power curves were fitted by 2$^{nd}$ order polynomial non-linear regression, and the maximal power production was determined from the apex of the force-power curve.

## Myosin heavy chain isoform determination and distribution

Sodium dodecyl sulfate polyacrylamide gel electrophoresis was used to determine the myosin isoform composition of the muscle lysates and single fibers as previously described [35]. The stacking gel contained a 4% acrylamide concentration (pH 6.7), and the separating gel contained 8% acrylamide (pH 8.7) with 30% glycerol (v/v). The gels were run for 24 h at 15 ˚C and a constant voltage of 275 V. Gels for whole muscle lysates were stained with Coomassie blue and single fiber gels were silver-stained. Gels were scanned and analyzed with ImageJ (v1.49, NIH, USA).

## Statistical analyses

Data are presented as mean ± SEM. GraphPad Prism 6 was used to calculate statistics. For statistical analysis one-way ANOVA, two-way ANOVA and the t-test with multiple testing corrections were used, as appropriate. For permeabilized single fibers study, Student's t-tests were performed on force-pCa curve (each pCa) and $pCa_{50}$ of each fiber type and genotype (OM vs. Vehicle; Figs 3 & 4). For the force-frequency curve of the intact whole muscle experiment, paired t-tests were used at each stimulation frequency (OM vs. Vehicle). Student's t-tests were also used for comparison of genotype (CON vs. cKO) in MHC type I proportion (Fig 1A) and relative force increase (Fig 5A) in intact whole muscle experiment. Repeated measure 2-way ANOVA and multiple post-hoc t-test using Sidak method were performed for half-frequency, Hill-coefficient (Fig 5B and 5C), and maximal power (Fig 6B) analysis. $p < 0.05$ was considered to be statistically significant.

## Results

### Fiber type composition of *Neb* cKO muscle and Dose-response of OM

It has been previously reported that nebulin deficiency leads to a shift in myosin heavy chain isoform distribution from fast type II to slow type I MHC[25]. In order to test this in the mice studied here, SDS-PAGE was performed on lysates obtained from muscles used in the mechanical studies. Representative MHC gels and analyzed results are shown in Fig 1A. The average proportion of type I MHC isoform in soleus muscle was 51±3% in control (CON) muscle and 91±1% in *Neb* cKO muscle. Thus the soleus muscle of *Neb* cKO muscle expresses predominately type I myosin. We also performed studies to establish the OM dosage for our experiments and measured the OM effect in intact *Neb* cKO soleus muscle activated with 20 Hz stimulation. As shown in Fig 1B, OM significantly increased force; this effect was seen already at 0.05 µM OM and followed a dose-response curve with an $EC_{50}$ of 0.6 µM.

The OM dose-response curve was also established for skinned fiber bundles dissected from soleus muscle of *Neb* cKO mice. (Note that these fiber bundles are fiber-type mixed and that studies with fiber-typed single fibers are described later in the Results section). Fiber bundles were sub-maximally activated by exogenous calcium at a range of OM concentrations (S1A Fig). All tested OM concentrations increased force production. Interestingly, the force increase reaches a plateau between 0.5 µM and 1.0 µM, and declined at a higher dose (S1A Fig). The half-maximal effect occurred at ~0.1 µM (the force decline at 10 µM precluded fitting the data to a standard dose-response curve). For a comparison we also performed experiments on sub-maximally-activated permeabilized left ventricular papillary muscle and found that force increased more gradually and did not decline at high OM doses (S1B Fig, see also Discussion).

We also determined the OM effect on the full force-pCa relationship of soleus fiber bundles. At 0.1 and 0.3 µM OM, force was increased, particularly at submaximal activation levels (Fig 2A and 2B). At 0.5 µM and higher the force increase at submaximal activation was small and at

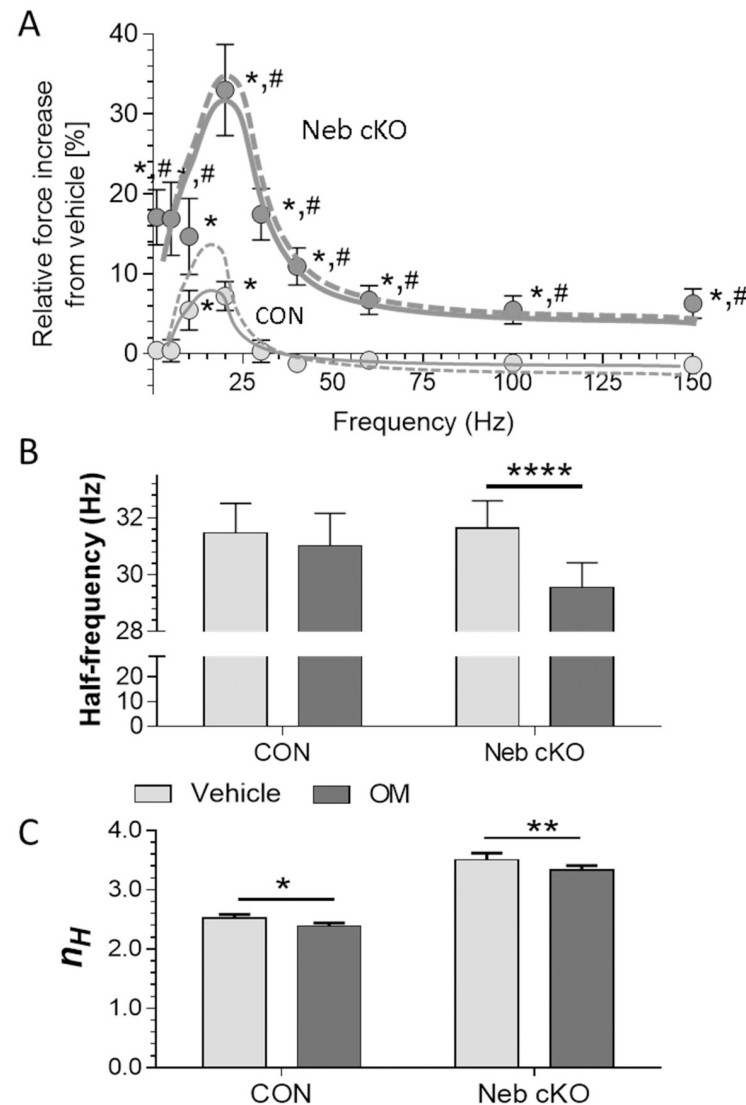

**Fig 5. Effect of OM on force of intact soleus muscle.** (A) Relative increase in force as a function of frequency in OM over vehicle. *Neb* cKO muscle (dark symbols); CON (light symbols). The relative increase in force with OM over vehicle was increased at all stimulation frequencies in *Neb* cKO muscles, while significance was found at only 10 and 20Hz in CON muscle. The OM effect was significantly greater in *Neb* cKO than CON at all frequencies except 10 Hz. The solid lines are the fits to the data and the broken lines normalize the response to the slow type I myosin content (50% in control muscle and 90% in *Neb* cKO muscle ([Fig 1A])). (B-C) frequency for half-maximal force (half-frequency) in (B) and Hill-coefficient in (C) with vehicle or OM. A two-way repeated measures ANOVA revealed in B) a significant treatment effect with a significant interaction with genotype and in C) a significant treatment effect, a significant genotype effect and an interaction between treatment and genotype. Sidak multiple comparison tests revealed in B) a significant OM effect in *Neb* cKO muscle and in C) a significant OM effect in both CON and *Neb* cKO muscles. The number of muscles is 14 for each genotype. Values are mean ± SEM. Asterisk(*) comparison of OM vs. vehicle; sharp(#) comparison cKO vs. CON. One through four symbols: p<0.05, p<0.01, p<0.001, and p<0.0001, respectively.

maximal activation, force was depressed ([Fig 2B]). Note that force depression was absent in the intact muscle experiments ([Fig 1]), most likely because these experiments were performed at a frequency (20 Hz) that produces submaximal force levels (see also [Discussion]). Normalized force-pCa curves of soleus fiber bundles, with force normalized to the maximal force (pCa

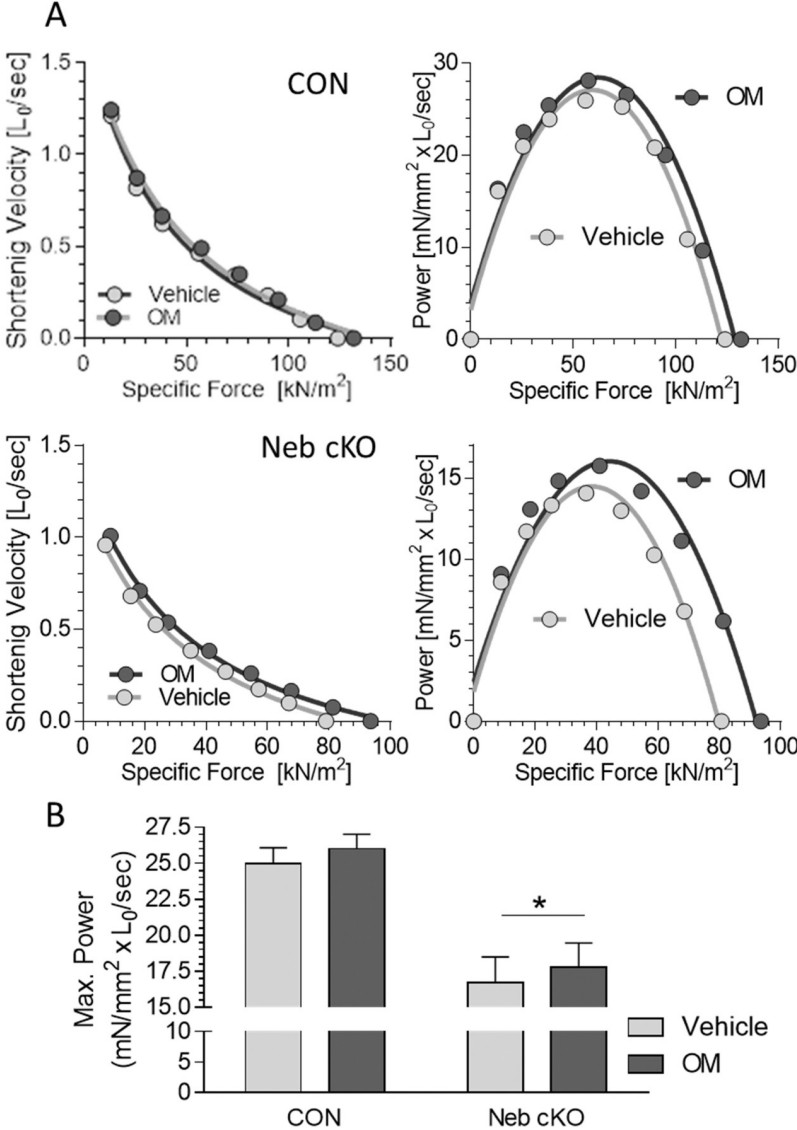

**Fig 6. Effect of OM on power production in intact soleus muscle.** A) Examples of force-velocity (left) and force-power curves (right) in CON (top) and *Neb* cKO (bottom) in vehicle and OM. (B) Maximal power with vehicle and OM solutions in both genotypes. A two-way repeated measures ANOVA revealed a significant treatment effect and a significant genotype effect. A Sidak multiple comparison test revealed a significant OM effect in *Neb* cKO muscle. *Neb* cKO mice have decreased power compared to CON. OM-treatment increased power in *Neb* cKO mice. Numbers of muscles are 6 for CON and 11 for cKO. Values are mean ± SEM. Comparison between vehicle and OM is marked as an asterisk(*). One symbol p<0.05.

4.0), are left shifted in OM (increased $pCa_{50}$) and less steep (reduced Hill coefficient), Fig 2C. We also investigated cross-bridge cycling kinetics by measuring the rate constant of tension redevelopment ($k_{tr}$) in *Neb* cKO bundles at a range of OM-concentrations. Starting at 0.5 µM OM, $k_{tr}$ is decreased (Fig 2D). As $k_{tr}$ correlates with myosin ATPase activity [34, 38], these experiments suggest that high OM doses slow ATPase activity. Based on these results and the desire to limit the likelihood of OM effects on cardiac muscle [18, 27, 28, 30] (see also Discussion), 0.1 µM OM was used in a follow-up study focused on single fibers that have been fiber-typed.

## Force measurements on fiber-typed single fibers

The effect of OM on contractility was first studied by measuring the $Ca^{2+}$ dependence of force in membrane-permeabilized soleus slow type I and fast type II single fibers. This comparative study was performed only on CON fibers (that express nebulin) because of the limited number of type II fibers in *Neb* cKO soleus muscle (out of the 50 studied fibers only four fibers (one treated with vehicle and three treated with OM) were type II, which precluded statistical testing). Examples of single fiber gels are shown in Fig 3A. Single fibers were activated with incrementally increasing concentrations of $Ca^{2+}$ with or without 0.1 μM OM. Treatment of fibers with OM resulted in a robust increase in submaximal force production in type I fibers (Fig 3B). We did not observe any effect of OM on type II fibers (Fig 3C). The results in type I fibers show an OM-induced increase in $pCa_{50}$ (the pCa for half-maximal force) of 0.29 pCa-units (insets of Fig 3B and 3C). Thus, at a submaximal activation level, OM significantly affects force production of type I fibers.

Soleus type I fibers from *Neb* cKO mice were studied next. Compared to type I fibers from CON muscle these fibers had significantly lower maximal specific force and significantly lower calcium sensitivity (Table 1), as shown by their reduced $pCa_{50}$ (6.12 in cKO and 6.36 in CON). The low specific-force levels of the *Neb* cKO fibers are likely due to the absence of nebulin but the role of other adaptive changes cannot be excluded. It is known that at the whole muscle level profound changes in expression occur in MHC, Troponin-T, Troponin-I and Tropomyosin isoforms [25]. This is likely due to the known fiber-type switch from fast to slow, but determining whether differences exist when comparing the same fiber type in different genotypes will require future work focused on single fiber proteomics. OM had no effect on the maximal active force, but at submaximal $Ca^{2+}$ levels, forces were significantly increased (Fig 4A). The greatest increase was at low $Ca^{2+}$ levels, the effect became less as $Ca^{2+}$ was increased and was absent at maximal activation (Fig 4A). At intermediate calcium levels the increase was large, for example at pCa 6.5, OM increased specific force from 8.4±2.6 $mN/mm^2$ to 19.2±3.5 $mN/mm^2$. However, it is important to point out that although specific force was more than double; it still was much below that of untreated control fibers (55 $mN/mm^2$ at pCa 6.5).

The OM-induced increase in force at sub-maximal activations in type I *Neb* cKO fibers resulted in a leftward shift of the force-pCa curves (Fig 4A). The Hill coefficient ($n_H$), a measurement of the filament cooperativity, was also determined. Although mean $n_H$ values were reduced by OM, the differences were not significant (Table 1). The OM-induced leftward shift of the force- pCa relation increased the $pCa_{50}$ from 6.12±0.10 to 6.46±0.08 in *Neb* cKO fibers (Fig 4A) which is close to that of type I CON fibers in vehicle (pCa 6.36, Table 1).

**Table 1. Effect of OM on active force of single fibers (top) and intact soleus muscle (bottom).**

| Single fibers | Max. Specific force ($mM/mm^2$) | | $pCa_{50}$ | | | $n_H$ | | |
|---|---|---|---|---|---|---|---|---|
| | Veh | OM | Veh | OM | Δ | Veh | OM | Δ |
| CON type II | 131.4±5.5 | 146.3±8.1 | 6.20±0.06 | 6.23±0.04 | 0.03±0.01 | 3.66±0.11 | 3.26±0.44 | 0.54±0.33 |
| CON type I | 130.5±4.56 | 144.1±5.60 | 6.36±0.08 | 6.65±0.05 (*) | 0.29±0.10 | 1.81±0.35 | 1.54±0.33 | 0.28±0.27 |
| *Neb* cKO type I | 43.6±3.16 (####) | 41.6±4.88 (####) | 6.12±0.10 (#) | 6.46±0.08 (*) | 0.34±0.04 | 2.10±0.31 | 1.84±0.52 | 0.36±0.62 |
| **Whole muscle** | Max. Specific force ($mM/mm^2$) | | Freq. for 50% of maximal force (Hz) | | | $n_H$ | | |
| | Veh | OM | Veh | OM | Δ | Veh | OM | Δ |
| CON | 297.8±15.1 | 294.1±15.7 | 31.5±1.0 | 31.0±1.1 | 0.45±0.27 | 2.52±0.06 | 2.39±0.05(*) | 0.13±0.02 |
| *Neb* cKO | 118.6±9.4 (####) | 124.6±8.9 (***,####) | 31.6±1.0 | 29.5±0.8 (****) | 2.10±0.27(###) | 3.50±0.11 (####) | 3.33±0.07 (*,####) | 0.17±0.06 |

(*) comparison between OM and Veh (#) comparison between *Neb* cKO and CON

## Effect of OM on intact muscle

Intact soleus muscles were electrically stimulated at a range of frequencies to determine their force-frequency relation. The maximal specific force of *Neb* cKO soleus was ~1/3 of the maximal specific force of CON soleus, consistent with the maximally activated single fiber data (Table 1) and highlighting that the force deficit of intact *Neb* cKO muscle has a myofilament origin. OM has a small but significant effect on force at all stimulation frequencies. The increase was maximal at 20Hz (32±5%) and gradually decreased at higher stimulation frequencies (Fig 5A, dark symbols). However, in CON muscle the effect was less (Fig 5A, light gray symbols) and only reached significance at 10 Hz (increase 5±2%) and 20Hz (increase 7±2%). From the force-frequency curves, the frequency for half-maximal force increase beyond twitch force (defined as half-frequency) and the Hill coefficient ($n_H$) were calculated (Fig 5B and 5C). The average half-frequency was decreased by OM but this effect was only significant in *Neb* cKO muscle (Fig 5B and Table 1). Hill coefficients were significantly decreased by OM in both genotypes, and the effect was largest in cKO muscle (Table 1).

The shortening velocity at a range of load levels was also measured. The shortening velocity was normalized by muscle length, and the force-velocity curves were converted to force-power curves and the effects of OM were studied. Fig 6A show example experiments and 6B show the mean results of all experiments. OM had no significant effect on CON muscles but it enhanced the maximal power in *Neb* cKO muscles, the effect was small but significant (Fig 6B).

## Discussion

Nemaline myopathy (NEM) is the most common non-dystrophic congenital myopathies, with features that include muscle weakness, muscle atrophy, presence of nemaline rod-bodies and type I fiber predominance[8, 9, 12–17]. It impacts both peripheral and respiratory muscles, and many NEM patients have difficulties in locomotion and respiratory function[17]. However, targeted treatment options for NEM patients are not available. Although OM has been developed as a heart failure drug, the expression of cardiac myosin (Myh7 or β-MHC) in both heart and slow skeletal muscle[21], predicts that OM is effective in slow skeletal muscle as well, and the findings of the present study clearly bear this out. OM greatly increases the force of single type I fibers activated at submaximal activation levels and significantly increases the calcium sensitivity, as reflected by the increased $pCa_{50}$ (Fig 4). Considering the large number of type I fibers in humans [22–24] and the additional shift in fiber type distribution of nebulin-deficient muscle and NEM patients toward type I fibers [12–15], OM might be a therapeutic option to ameliorate the muscle weakness in NEM patients. That OM is well-tolerated and is currently in phase-3 clinical trials for treatment of heart failure makes it an attractive therapeutic option. Our findings suggest that OM is far from a cure. However, it has a good potential to bring relief in nebulin-based nemaline myopathy patients, not only because of their type I predominance but also because nebulin deficiency appears to increase the effectiveness of OM. Below we discuss our findings in detail.

OM has been developed to increase the contractile force of cardiac muscle in heart failure patients[18]. Unlike various existing calcium sensitizers[39] OM does not affect calcium transients, which limits possible side effects (e.g., altering calcium-dependent signaling processes) [18, 40]. Recent single molecule studies [20] provide evidence that OM binds myosin, causing long-lived strongly bound myosin heads that do not generate force. From these studies, a model was proposed in which these OM-bound heads cooperatively activate the thin filament at submaximal activation levels, recruiting OM-free myosin heads and thereby increasing force [20]. This mechanism naturally explains the sarcomere length dependence of the magnitude of the OM-effect that has been found in sub-maximally activated cardiac muscle[41],

with larger OM effects at short sarcomere length where the baseline activation level of the thin filament is lower and more OM-free heads can be recruited to the force generating pool than at longer sarcomere length. It can also explain the reduced OM effects at high OM levels reported by Nagy et al. for both cardiac myocytes and diaphragm fibers[42], and seen here in soleus fiber bundles (S1A Fig): at high OM levels the OM-bound heads that do not generate force outnumber the additionally recruited OM-free force generating heads. The dose-response curve that we measured on cardiac muscle does not decline at high OM levels (S1B Fig) a finding similar to that of Gollapudi et al (their Fig 2)[41]. This can be explained by the short sarcomere length that was used in these studies (2.0 μm in our study and 1.9 μm in [41]), where baseline activation is low, and the long sarcomere length used by Nagy et al in their cardiac myocyte study (2.3 μm) where the baseline activation level is high. Thus a mechanism based on an increased attachment lifetime of OM bound non-force producing myosin heads that enhances thin filament activation can explain the various findings. An additional mechanism recently proposed by Kampourakis et al. [43] to explain the OM effect involves the ON and OFF states of the thick filament [44, 45]. In the OFF state, many of the myosin heads in the thick filament are bent back towards the center of the sarcomere and are unable to interact with actin and in the ON state they are in a more perpendicular position and available for actin binding and force generation. By promoting the ON state of the thick filament and increasing the thin filament activation level, OM might facilitate actomyosin interaction and enhance force development at low to intermediate calcium levels [43, 45, 46]. The extent to which both thick filament and thin filament based mechanisms are responsible for OM-induced force increase requires future research.

The effects that were found in type I fibers (increased $pCa_{50}$-values and no effect on the maximal force) are in general similar to the OM effects reported in cardiac muscle. In cardiac muscle of mouse[47], rat[42, 43], guinea pig[41] and human[28], OM increases calcium sensitivity (higher $pCa_{50}$), and typically has no effect on the maximal active force. Our findings are also consistent with the work of Nagy et al who recently used OM on rat diaphragm muscle fibers[42]. Although the myosin isoform was not directly determined, Nagy et al did measure the kinetics of force generation ($K_{tr}$) which also provides insights in the fiber type (slow kinetics for type I fibers and fast kinetics for type II fibers). In both their and our study, OM clearly increases the calcium sensitivity of slow type I fibers (fibers with low $K_{tr}$ values in Nagy et al [42] and Myh7 in our study). The Nagy et al study also found a small but significant effect of OM on type II fibers (identified by Nagy et al as having fast kinetics[42]), which we did not observe. This issue warrants follow-up. If future studies were to show that OM also affects type II MHC isoforms, it would extend OM's usefulness for treating skeletal muscle diseases with as it would increase the proportion of fibers being affected by OM. It would also represent a treatment option for myopathies with fiber type II dominance.

Most cardiac studies were performed at OM levels higher than ours (0.3–1.0 μM), except for studies on guinea pig myocardium[41] and human myocardium[28] (both of which found no effect at 0.1 μM) and a study on rat myocytes [45]that did find an effect at 0.1 μM but that was smaller than in the present study (e.g., $\Delta pCa_{50}$ ~0.1, our study ~0.3). Thus it appears that the sensitivity to OM is higher in our study on type I skeletal muscle fibers than in published studies on cardiac muscle. The explanation for this is not immediately clear but could reside in the distinct protein isoform composition of thin and thick filament proteins involved in activation in slow skeletal muscle compared to cardiac muscle. It is also possible that the tissue-specific expression of nebulin (present in skeletal muscle but absent in cardiac muscle[48]) is part of the explanation, or that some other putative adaptation in the proteome of nebulin-free skeletal muscle fibers play a role. Future work is required to fully understand the details of the OM effect on both cardiac and skeletal muscle. The higher OM sensitivity of skeletal muscle type I

fibers, compared to cardiac muscle, is beneficial as it provides a therapeutic window for a sole skeletal muscle effect. For OM to be useful for increasing skeletal muscle force in nebulin-based nemaline myopathy patients, undesirable effects in the heart have to be avoided. Slowing of force relaxation in the heart is of a particular concern since this can impair diastolic filling and cause ischemia, especially at high heart rates. However, clinical trials have shown that OM is well tolerated without adverse effects up to ~10 μM in healthy individuals, and in heart failure patients at plasma concentrations of up to ~1.0 μM (reviewed in[39]). Thus it is possible that the doses established in the present study will be able to increase skeletal muscle force in Nemaline Myopathy patients without adverse effects on the heart. Finally, it is also worth pointing out that when using OM in heart failure patients, undesirable effects on skeletal muscle have to be avoided. This could occur, for example, if high OM doses required for cardiac benefits in heart failure patients were to suppress force production or crossbridge cycling kinetics of type I fibers of the diaphragm. The possible clinical application of OM requires study of both the cardiac and skeletal muscle systems.

The effect of OM on intact muscle reached a maximum at 20Hz and then gradually decreased at higher stimulation frequencies (Fig 5A). This is similar to the findings in single fibers and, thus, in both preparations, the greatest effect of OM occurs at submaximal activation levels. Because skeletal muscles operate at submaximal activation levels during normal activity [22, 49, 50], OM is expected to enhance force development under clinically meaningful conditions. OM does not nearly recover the large functional deficit of nebulin deficient muscle and, thus, OM is far from 'a cure.' Considering that our work was performed at a low OM concentration (0.1 μM), which is below the $EC_{50}$ of 0.6 μM, it is likely that the effect of OM can be augmented by increasing the OM dose. From our present study it is possible to conclude OM is likely to provide relief to NEM patients as increases in force of ~20% are clearly attainable, which is likely to improve quality of life for patients with muscle weakness.

It is also worth noting that OM has a small effect on intact muscle of *Neb* cKO mice at simulation frequencies that result in maximal tetanic force (Fig 5A), and that this is not the case in single fibers that are maximally activated (pCa 4.0) (Fig 4B, Table 1). This suggests that unlike in maximally activated skinned fibers, in intact muscle of *Neb* cKO mice, the number of myosin molecules participating in maximal tetanic contraction is not at its maximal level. This conclusion is consistent with an X-ray study on intact muscle that revealed that, compared to CON muscle, *Neb* cKO muscle has thick filaments that partially remain in the OFF state during a maximal tetanus [51]. Our findings indicate that OM corrects this effect by more completely switching the thick filament to its ON state. This mechanism (a deficit in the ON state of the thick filament in the absence of nebulin) can also explain the larger effect of OM on type I fibers of *Neb* cKO muscle compared to CON type I fibers. Finally, it also provides an explanation for the only study in which OM was found to increase the force of maximally activate skinned cardiac muscle (pCa ~4.0)[41]. This finding was obtained at a short sarcomere length (1.9 μm) where the thick filament might not be fully turned ON with calcium[52], and, thus, OM is able to augment the ON state of the thick filament and increase maximal force.

In summary, this study shows for the first time that at submaximal activation levels OM increases force and power of slow skeletal muscles. This is expected to be beneficial for nemaline myopathy patients where type I fibers dominate, with predicted benefits for respiratory and peripheral muscle function. OM currently is in a Phase 3 Heart Failure clinical trial (GALACTIC-HF). The previous phase I and phase II clinical trials have shown that OM is well tolerated to plasma levels of ~1.0 μM [39], a level beyond the level used in the present study. We conclude that OM has the potential to improve skeletal muscle function in NEM patients and bring much needed relief. Finally, it is also worth considering that although type II fibers do not appear to benefit from OM, fast skeletal muscle troponin activators (FASTA) have been

developed that selectively activate type II fibers by increasing their sensitivity to calcium [53]. Indeed it has been shown that FASTAs increase force at submaximal activation levels in type II skeletal muscle fibers from nebulin-deficient mice and nemaline myopathy patients [54, 55]. Thus, by using a combination of both OM and FASTA, an increase in force of all fiber types might be achievable.

## Supporting information

**S1 Fig.** A) Dose-response of OM effect on force production by soleus fiber bundles from Neb cKO mice (n = 6). Specific force was measured at pCa 6.75. OM increases specific force at low OM doses, the effect is maximal between 0.5 and 1.0 μM OM, and is much less at 10 μM OM. B) OM Dose-response using LV permeabilized papillary muscle from control (n = 7) and Neb cKO mice (n = 7). Specific force was measured at pCa 6.0. Specific force follows a dose-response curve with EC50 of 0.62±0.04 μM OM (control) and 0.79±0.08 μM OM (Neb cKO). No significant difference in EC50 (See text for details).
(TIF)

## Acknowledgments

We are grateful to Dr. Coen Ottenheijm, Ms. Luan Wyly, Mr. Chandra Saripalli, Ms. Xiaoqun Zhou, Mr. Xiangdang Liu for important support of this work.

## Author Contributions

**Conceptualization:** Henk Granzier.

**Data curation:** Johan Lindqvist, Eun-Jeong Lee, Esmat Karimi, Justin Kolb.

**Formal analysis:** Johan Lindqvist, Eun-Jeong Lee, Esmat Karimi, Justin Kolb.

**Funding acquisition:** Henk Granzier.

**Investigation:** Eun-Jeong Lee, Justin Kolb, Henk Granzier.

**Supervision:** Henk Granzier.

**Writing – original draft:** Johan Lindqvist, Eun-Jeong Lee, Henk Granzier.

**Writing – review & editing:** Justin Kolb, Henk Granzier.

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
