## [Decision Letter · Decision Letter 0]

6 Aug 2019

PONE-D-19-14908

Omecamtiv mecarbil lowers the contractile deficit in a mouse model of nebulin-based nemaline myopathy.

PLOS ONE

Dear Dr. Granzier,

Thank you for submitting your manuscript to PLOS ONE. After careful consideration, we feel that it has merit but does not fully meet PLOS ONE’s publication criteria as it currently stands. Therefore, we invite you to submit a revised version of the manuscript that addresses the points raised during the review process.

Please provide a detail response to the queries submitted by reviewers and keep in mind that there are new guidelines for making figures involving blots and gels that you should consider when making the final figure.

We would appreciate receiving your revised manuscript by Sep 20 2019 11:59PM. To enhance the reproducibility of your results, we recommend that if applicable you deposit your laboratory protocols in protocols.io, where a protocol can be assigned its own identifier (DOI) such that it can be cited independently in the future. For instructions see: http://journals.plos.org/plosone/s/submission-guidelines#loc-laboratory-protocols

We look forward to receiving your revised manuscript.

Kind regards,

Agustín Guerrero-Hernandez

Academic Editor

PLOS ONE

2. To comply with PLOS ONE submissions requirements, in your Methods section, please provide additional information on the animal research and ensure you have included details on (1) methods of sacrifice for all parts of your study (including the study described in section 2.3), (2) methods of anesthesia used prior to cervical dislocation.

Reviewers' comments:

Reviewer's Responses to Questions

**Comments to the Author**

1. Is the manuscript technically sound, and do the data support the conclusions?

Reviewer #1: Yes

Reviewer #2: Yes

2. Has the statistical analysis been performed appropriately and rigorously? 

Reviewer #1: I Don't Know

Reviewer #2: Yes

3. Have the authors made all data underlying the findings in their manuscript fully available?

Reviewer #1: No

Reviewer #2: Yes

4. Is the manuscript presented in an intelligible fashion and written in standard English?

Reviewer #1: Yes

Reviewer #2: Yes

5. Review Comments to the Author

Reviewer #1: This paper describes the effects OM has on a nebulin KO model. The premise is that since type 1 fibres are mainly affected and predominate in nemaline myopathy due to nebulin mutations, it may have a positive effect on contractility. Whilst 0.1µM OM has limited effects that are greater in the nebula KO, there is a considerable issue about the chosen dosage.

1 In this system EC50 appears to be about 0.6µM (no sem is given), so 0.1µM is a rather low level of dosage. It is necessary to explore this system in greater depth, using-

1a Effects on contractility and pCa50 at a range of doses- is there a maximum with decline at higher concentrations as Nagy et al. found?

1b The low dose is chosen to avoid cardiac effects. Since none of the experiments involves cardiac muscle this is not relevant to the study. However it would be important to check the dose-response curve of cardiac muscle in this system and also of the nebulin KO, which may be different from wt.

1c There should be some discussion as to whether a combined cardiac and type 1 fibre effect might actually be advantageous given that nebulin is also present in heart muscle.

2 The authors seem to have misunderstood the mechanism of OM action. (line 46) no mention of the work of Woody et al (ref 36) which clearly explains the mechanism- in fact refs 21 and 22 seem to be wholly unsuitable, being clinically al trials.

In the discussion lines 310-317 are essentially wrong- Woody shows that OM causes non-moving attached cross bridges that are inhibitory but which activate the THIN FILAMENT cooperatively. SRX has nothing to do with this- you are thinking of Mavacamten, perhaps.

3 in the discussion the authors should compare OM with the fast skeletal-specific troponin activators as treatments for Nemaline myopathy. Some of these were tested with the nebulin KO mouse.

(Collibee et al. 2018; Hwee et al. 2017, 2015,2014; de Winter et al. 2013

4 Figures. You should use a consistent shade for the points on graphs: in figs 2 & 3 OM is light, in Figs 4 & 5 OM is dark !

other points:

Some poorly constructed sentences; the text should be inspected and corrected by an English expert. line 15 is an obvious example.

line 37 nebulin is BELIEVED to play.....

line 41 Few or none? If a few state what they are.

line 53 Can you specify which are the main type 1 muscles (in mouse and in human)

Figure 5- Please also show the original force-velocity curves which were used to create the force-power plots.

Reviewer #2: This MS by Lee et al reports a study of the effect of omecamtiv mecarbil (OM), a small molecule activator of cardiac beta/skeletal muscle type I myosin, on correcting the contractile abnormality of mouse skeletal muscle lacking nebulin for potential use in the treatment of NEB nemaline myopathy. Skinned single fiber and intact muscle studies were performed. The results suggest beneficial effect of OM treatment. The studies used comprehensive quantitative approaches which the authors have expertise and extensive past experiences. The experiments are carefully performed and the data are clearly presented. While this work is considered as potentially interesting to the field by providing valuable information on whether and how manipulate myosin ATPase function and kinetics could therapeutically correct or mitigate the functional defect of a thin filament function due to the loss of nebulin, a skeletal muscle specific protein. There are some issues that need the authors’ attention:

1. For the rationale and significance of this study, the authors should emphasize the effect of OM on increasing myosin activity to treat myopathies, since this drug is unlikely to be directly useful in the treatment of Neb myopathy for its cardiac effect in humans.

2. Muscle weight and fiber size information should be presented and discussed for whether and how atrophy has effect on the contractile function of whole muscle or single fibers.

3. The whole muscle functional parameters should be normalized to the contents/% of type I myosin.

4. Since there was no anticipated effect of OM on maximum force whereas it increased calcium sensitivity (pCa50), the effect on intrinsic myosin ATPase activity should evaluated. The authors referenced previous Ktr study, which should also be examined here.

5. In addition to Force-Power curve, the primary contractile velocity data should be shown and discussed.

6. Examples of the SDS-gel confirmation of single fibber types should be shown. This will also help to evaluate the myofilament protein contents of the Neb KO and WT muscle fibers studied. If there are any adaptive changes in the KO muscle, they can be discussed for potential significance in altering contractility.

7. The difference between normal cardiac and skeletal muscles in their nebulin contents may contribute to their contractile properties and the responses to OM treatment, a worthwhile point for discussion.

6. PLOS authors have the option to publish the peer review history of their article (what does this mean?). If published, this will include your full peer review and any attached files.

Reviewer #1: No

Reviewer #2: No

---

## [Author Response · Author response to Decision Letter 0]

1 Oct 2019

Reviewer #1: 

Comment: “This paper describes the effects OM has on a nebulin KO model. The premise is that since type 1 fibres are mainly affected and predominate in nemaline myopathy due to nebulin mutations, it may have a positive effect on contractility. Whilst 0.1µM OM has limited effects that are greater in the nebula KO, there is a considerable issue about the chosen dosage.

1a Effects on contractility and pCa50 at a range of doses- is there a maximum with decline at higher concentrations as Nagy et al. found?

1b The low dose is chosen to avoid cardiac effects. Since none of the experiments involves cardiac muscle this is not relevant to the study. However it would be important to check the dose-response curve of cardiac muscle in this system and also of the nebulin KO, which may be different from wt.

 1c There should be some discussion as to whether a combined cardiac and type 1 fibre effect might actually be advantageous given that nebulin is also present in heart muscle.”

Response: First of all, thank you reviewer for the positive overall evaluation and the constructive comments that allowed us to strengthen our manuscript. We have taken your comments to heart and performed the following experiments. 

To answer the issue about the maximum with decline at higher concentrations (question 1a) we performed a new series of experiments on Neb cKO fiber bundles that were activated at a range of pCa values and in the presence of 0, 0.1, 0.3, 0.5, 1.0 and 10 µM OM (new Figure 2, text on page 12, lines 215-222). At submaximal activation OM increased force and the effect peaked at 0.3 µM OM and declined at OM >0.5 µM OM. Similar findings are reported by Nagy et al. who studied permeabilized rat diaphragm fiber bundles activated to ~25% of maximal. They found that OM increased force with a maximum at ~0.3 µM OM and a reduction in the effect at OM >~1.0 µM OM. These new studies support the earlier conclusion that Neb cKO fibers 0.1 µM OM is sufficient to see robust effects on force at submaximal activation. Although 0.3 µM OM has a slightly larger effect, the benefit beyond 0.1 µM is small and to limit the potential for cardiac effects we elected to use in our single fiber experiments 0.1µM OM. 

In response to question 1b we measured the dose-response curve of cardiac muscle in both WT and Neb cKO LV papillary muscle (See Supplemental Figure 1B). We find EC50 values of ~0.7 µM, similar to Nagy et al for rat cardiac myocytes, but we find no drop-off at OM concertation >1.0 µm, unlike in the work of Nagy et al (their Fig. 1). Please note that other studies (e.g., Gollapudi et al, Biophys J, 2017) also found no drop-off. We believe that this is likely due to a sarcomere length difference (we worked at 2.0 µm SL as did Gollapudi et al, whereas Nagy et al, used 2.3 µm, a length at which the baseline activation level is already high due to length-dependent activation); future follow-up work is required to test this and other possible explanations. In our revised manuscript we discuss the issue of ‘drop-off ‘that that is sometimes present but not always. See page 20, line 368-376.

In response to question 1c, we would like to first highlight that nebulin is not expressed in adult cardiac muscle (Kolb et al., J Mol Cell Cardiol. 2016;97:286-94.). As for the possible utility of omecamtiv in patients, this has to be evaluated using an organismal-level perspective. For omecamtiv to be useful for increasing force of skeletal muscle in nebulin-based nemaline myopathy patients, undesirable effects in the heart have to be avoided. It is known that at high OM concentrations, the relaxation rate of the heart is reduced and that this can impair diastolic filling and cause ischemia, especially at high heart rates. This has to be avoided when treating skeletal muscle myopathies with OM. Fortunately, these cardiac effects occur at high OM concentrations (>~ 1 µM), or ~10-fold higher concentrations than those that are effective in increasing skeletal muscle force. We now add this as a discussion item, see page 22 lines 423-35. 

Comment: “2 The authors seem to have misunderstood the mechanism of OM action. (line 46) no mention of the work of Woody et al (ref 36) which clearly explains the mechanism- in fact refs 21 and 22 seem to be wholly unsuitable, being clinically al trials.”

Response: We have updated our discussion of the OM mechanism. See page 3, lines 45-50.

Comment: “In the discussion lines 310-317 are essentially wrong- Woody shows that OM causes non-moving attached cross bridges that are inhibitory but which activate the THIN FILAMENT cooperatively. SRX has nothing to do with this- you are thinking of Mavacamten, perhaps.)”

Response: We had referred to the work by Kampourakis et al (J. Physiol, 2018) who used omecamtiv (not mavacamten) to investigate the thick filament ‘on’ and ‘off’ states. These authors provide evidence that in passive muscle OM promotes the thick filament ON state. Although the Woody et al study deserves most of the focus in our Discussion, we don’t think that this study excludes that in addition to enhancing thin filament activation, OM might also increase the thick filament ON state. In response to your comments we have revised our Discussion. See page 20, lines 386-395

Comment: “3 in the discussion the authors should compare OM with the fast skeletal-specific troponin activators as treatments for Nemaline myopathy. Some of these were tested with the nebulin KO mouse. (Collibee et al. 2018; Hwee et al. 2017, 2015,2014; de Winter et al. 2013)”

Response: We have added this suggested discussion item. See page 24, lines 469-475.

Comment: “4 Figures. You should use a consistent shade for the points on graphs: in figs 2 & 3 OM is light, in Figs 4 & 5 OM is dark !)

Response: You are right and we have made the suggested change. See revised Figures 3 and 4.

Comment: “Some poorly constructed sentences; the text should be inspected and corrected by an English expert. line 15 is an obvious example.

Responses: Done. See revised ms. and throughout.

Comment: line 37 nebulin is BELIEVED to play.....

Responses: Indeed this is better and we made the suggested change. See line 37 of revised ms.

Comment: line 41 Few or none? If a few state what they are.

Responses: Done. See line 42 of revised ms.

Comment: line 53 Can you specify which are the main type 1 muscles (in mouse and in human)

Responses: Done. See top of page 3, lines 54-56.

Comment: Figure 5- Please also show the original force-velocity curves which were used to create the force-power plots.”

Responses: Done. See left panels of Fig. 6A

Reviewer #2

“This MS by Lee et al reports a study of the effect of omecamtiv mecarbil (OM), a small molecule activator of cardiac beta/skeletal muscle type I myosin, on correcting the contractile abnormality of mouse skeletal muscle lacking nebulin for potential use in the treatment of NEB nemaline myopathy. Skinned single fiber and intact muscle studies were performed. The results suggest beneficial effect of OM treatment. The studies used comprehensive quantitative approaches which the authors have expertise and extensive past experiences. The experiments are carefully performed and the data are clearly presented. While this work is considered as potentially interesting to the field by providing valuable information on whether and how manipulate myosin ATPase function and kinetics could therapeutically correct or mitigate the functional defect of a thin filament function due to the loss of nebulin, a skeletal muscle specific protein.”

Responses: Thank you for your positive evaluation and for your comments that further helped us to improve our manuscript. 

 There are some issues that need the authors’ attention:

Comment: “1. For the rationale and significance of this study, the authors should emphasize the effect of OM on increasing myosin activity to treat myopathies, since this drug is unlikely to be directly useful in the treatment of Neb myopathy for its cardiac effect in humans.”

Responses: It is possible that it will turn out that OM will not be useful for nebulin-based nemaline myopathy patients but, in our opinion, the jury is still out and more future work will be required. Based on clinical studies ~1 µM OM is well tolerated in patients (see manuscript for details). Thus the 10-fold lower dose that is effective in type-I skeletal muscle fibers should be well-tolerated. Even if it turns out that OM will not be useful for treating skeletal muscle myopathies, the effect of OM on type-I skeletal muscle fibers still needs to be carefully considered, because skeletal muscle effects will have to be taken into account when treating heart failure patients with OM (e.g. depressing force in type-I diaphragm fibers has to be avoided). We added the suggested discussion, see page 22, lines 423-435.

Comment: “2. Muscle weight and fiber size information should be presented and discussed for whether and how atrophy has effect on the contractile function of whole muscle or single fibers.”

Responses: We have added this information, see page 5, lines 71-76 and page 6, lines 104-105. Please note that when forces are compared between genotypes, specific force is shown (force normalized to the cross-sectional area of the fibers/muscles), and muscle hypertrophy will not affect these comparisons. 

Comment: “3. The whole muscle functional parameters should be normalized to the contents/% of type I myosin.”

Responses: If effects were to occur at maximal activation, they could easily be normalized because all fiber types will be maximally activated under those conditions and their forces are additive. However, the effects of OM are mainly seen at submaximal activation and under those conditions the different fiber types will contribute to overall muscle force at different levels (because of their different force-frequency relations) adding uncertainty as to how to normalize by MHC content. Nevertheless, assuming simple addition at submaximal activation we have normalized for type-I MHC content (See Figure 5A of revised ms. and text on page 17, lines 323-325). 

Comment: “4. Since there was no anticipated effect of OM on maximum force whereas it increased calcium sensitivity (pCa50), the effect on intrinsic myosin ATPase activity should evaluated. The authors referenced previous Ktr study, which should also be examined here.”

Responses: Unfortunately we are currently unable to perform ATPase measurements in loaded fibers since we lost this expertise in our group and these measurements have to wait until some future opportunity arises to carry out this work. However, we have performed Ktr experiments and found that Kts is reduced at 0.5 μM OM and higher concentrations indicating decreased myosin ATPase activity, see new Figure 2C and page 13, line 233-237. 

Comment: “5. In addition to Force-Power curve, the primary contractile velocity data should be shown and discussed.”

Responses: Done. See Figure 6, left panels.

Comment: “6. Examples of the SDS-gel confirmation of single fibber types should be shown. This will also help to evaluate the myofilament protein contents of the Neb KO and WT muscle fibers studied. If there are any adaptive changes in the KO muscle, they can be discussed for potential significance in altering contractility.”

Responses: As you suggest, we now show gel examples of single fibers, see new Fig. 3A. As far as evaluating the protein composition of muscle fibers, this is easily doable for MHC because of the abundance of this protein, but for more typical proteins with lower abundance, this is challenging. Perhaps you meant studies at the whole muscle level. If so, we would like to point out that they have been done previously. We have studied the thin filament regulatory proteins (fast and slow skeletal muscle isoforms of Tm, TnI, TnT, and TnC) in soleus muscle and found the isoform composition to have shifted away from fast towards slow isoforms of skeletal muscle, when comparing Neb cKO and WT mice. This work has been published (Kiss et al., PNAS, 2018.). Performing this type of study at the single fiber level is challenging and time-consuming, in part due to the small size of mouse muscle fibers and especially in the Neb cKO mouse (see our previous work, Li et al, 2015, Human Molecular Genetics). For now, we revised our text highlighting our previous study at the whole muscle level and indicating that future work should focus on single fiber proteomics to reveal possible adaptive changes in the Neb cKO fibers (comparing the same fibers type in Neb cKO and WT mice). The difference between control and Neb cKO fibers is likely largely due to nebulin but adaptive changes (in addition to the fiber type switch) cannot be excluded and your point is well taken. See page 15, lines 275-281.

Comment: “7. The difference between normal cardiac and skeletal muscles in their nebulin contents may contribute to their contractile properties and the responses to OM treatment, a worthwhile point for discussion.”

Responses: We agree and now include this in the discussion. See page 22, lines 417-420. 

A final thank you to both reviewers for their time spent reviewing and for their excellent comments. We have taken these comments to heart, performed multiple additional experiments, and revised our manuscript throughout. We hope that you agree that our manuscript has been much improved and that our manuscript is now ready for acceptance. Thank you again.

---

## [Decision Letter · Decision Letter 1]

15 Oct 2019

Omecamtiv mecarbil lowers the contractile deficit in a mouse model of nebulin-based nemaline myopathy.

PONE-D-19-14908R1

Dear Dr. Granzier,

We are pleased to inform you that your manuscript has been judged scientifically suitable for publication and will be formally accepted for publication once it complies with all outstanding technical requirements.

With kind regards,

Agustín Guerrero-Hernandez

Academic Editor

PLOS ONE

Additional Editor Comments (optional):

Reviewers' comments:

Reviewer's Responses to Questions

**Comments to the Author**

1. If the authors have adequately addressed your comments raised in a previous round of review and you feel that this manuscript is now acceptable for publication, you may indicate that here to bypass the “Comments to the Author” section, enter your conflict of interest statement in the “Confidential to Editor” section, and submit your "Accept" recommendation.

Reviewer #1: All comments have been addressed

Reviewer #2: All comments have been addressed

2. Is the manuscript technically sound, and do the data support the conclusions?

Reviewer #1: Yes

Reviewer #2: Yes

3. Has the statistical analysis been performed appropriately and rigorously? 

Reviewer #1: Yes

Reviewer #2: Yes

4. Have the authors made all data underlying the findings in their manuscript fully available?

Reviewer #1: Yes

Reviewer #2: Yes

5. Is the manuscript presented in an intelligible fashion and written in standard English?

Reviewer #1: Yes

Reviewer #2: Yes

6. Review Comments to the Author

Reviewer #1: OK OK K OKK OKK OKK OKK OKK OKK OKK OKK OKK OKK OKK OKK OKK OKK OKK OKK OKK OKK OKK OK

Reviewer #2: My comments have been clearly addressed. I have no further concerns. This work should be published to make a valuable contribution to the field.

7. PLOS authors have the option to publish the peer review history of their article (what does this mean?). If published, this will include your full peer review and any attached files.

Reviewer #1: Yes: Steen Marston

Reviewer #2: Yes: J.-P. Jin

---

## [Editor Report · Acceptance letter]

5 Nov 2019

PONE-D-19-14908R1 

Omecamtiv mecarbil lowers the contractile deficit in a mouse model of nebulin-based nemaline myopathy. 

Dear Dr. Granzier:

I am pleased to inform you that your manuscript has been deemed suitable for publication in PLOS ONE. Congratulations! Your manuscript is now with our production department. 

With kind regards,

on behalf of

Dr. Agustín Guerrero-Hernandez 

Academic Editor

PLOS ONE